# Determinants of social participation in people with disability

**Tugce Pasin**☯, **Bilinc Dogruoz Karatekin**[ID]*☯

Department of Physical Medicine and Rehabilitation, Goztepe Prof Dr Suleyman Yalcin City Hospital, Istanbul, Turkey

☯ These authors contributed equally to this work.

* bilincdogruoz@hotmail.com

## Abstract

### Purpose

In this study, it is aimed to determine personal wellbeing and social participation levels across different physical disability types and levels of mobility.

### Methods

A sample of 85 individuals with physical disabilities, excluding those with mental disabilities were included. Sociodemographics, mobility of the participants, cause, duration of disability were recorded. Personal Wellbeing Index-Adult (PWI-A) scale was used for the assessment of wellbeing and Keele Assessment of Participation (KAP) for social participation.

### Results

Female, single, unemployed subjects and individuals with neurologic disability showed significantly higher median KAP-scores(p = 0.009, p = 0.050, p<0.001, p = 0.050, respectively). The median KAP-score of the independently mobile group was significantly lower compared to the other two groups (p = 0.001). The factors affecting KAP were determined as employment, mobility level and personal wellbeing (p = 0.002, p = 0.024, p = 0.050, respectively).

### Conclusion

Mobility level, employment and personal wellbeing are the determinants of social participation in people with disabilities. Neurological disability, female gender, being single, unemployment and mobility limitations are factors that reduce social participation.

## Introduction

Disability is the state of being unable to fulfill a role that is normal (depending on age, gender, and social and cultural factors) for an individual, due to an impairment [1]. It is an umbrella term for impairments, activity limitations, and participation restrictions; the term denotes the negative aspects of the interaction between an individual (with a health condition) and that

**Data Availability Statement:** Data are uploaded as supporting file.

**Funding:** The author(s) received no specific funding for this work.

**Competing interests:** The authors have declared that no competing interests exist.

individual's contextual factors (environmental and personal factors) as defined by the World Health Organization, International Classification of Functioning, Disability, and Health [2].

Besides the physical condition of the disabled individual, their psychosocial wellbeing and economic status also affect quality of life, which decreases as the dependency level of the person increases [3]. This underscores the multifaceted nature of disability, highlighting the need for holistic approaches that address not only functional limitations but also broader aspects of wellbeing and social integration.

Participation is a person's involvement in life situations and represents a crucial aspect of functioning and well-being [1]. Participation is vital for all people, especially for persons with disabilities. Participation not only improves physical health, but also provides emotional, psychological, social and societal benefits. Studies explaining the value of participation also underscored the fact that it has a significant impact on wellbeing and quality of life of people with disabilities [4]. Improved social participation and health status has been shown in people with disabilities in association with increased physical activity level or social interaction. In addition, it has been reported that functional limitation negatively affects social participation [5].

Determining the factors associated with participation in disabilities is of paramount importance for several reasons. Firstly, it allows for a deeper understanding of the complex interplay between individual characteristics, environmental factors, and personal experiences that influence a person's ability to engage meaningfully in various life activities. This understanding is crucial for developing targeted interventions and support services that can enhance participation and improve overall quality of life for individuals with disabilities. by unraveling the determinants of participation, we move closer to achieving a society that values diversity, respects individual differences, and ensures that everyone, regardless of their abilities or disabilities, has the opportunity to lead a fulfilling and inclusive life.

Despite the recognized importance of participation, there remains a need to delve deeper into the factors that influence social participation among individuals with disabilities. This study seeks to address this gap by examining and comparing key factors such as education, accomodation, employment, economic status, personal well-being, and social participation levels across different disability types and levels of mobility.

Therefore this study represents a significant step towards understanding the complex dynamics of social participation among individuals with disabilities. By examining a range of factors and their interrelationships, the study aims to contribute valuable insights that can inform interventions, policies, and practices aimed at promoting greater social inclusion and well-being for persons with disabilities.

## Materials and methods

### Sample selection

This study employed a descriptive, cross-sectional design and included individuals aged between 18 and 65 years who presented with physical disabilities for various reasons. Disability is an umbrella term for impairments, activity limitations, and participation restrictions; the term denotes the negative aspects of the interaction between an individual (with a health condition) and that individual's contextual factors (environmental and personal factors) as defined by the World Health Organization, International Classification of Functioning, Disability, and Health [2]. Individuals with physical disabilities were assessed at the physical medicine and rehabilitation outpatient clinic by the Health Board for Persons with Disabilities of a university hospital during a 3-month period from September 1 to December 1, 2021. The exclusion criteria were individuals with mental or cognitive disabilities and those who did not agree to participate in the study.

The sample size has been determined for the PWI variable. To work with a 10% margin of error at a 95% confidence level, considering a population mean of 66.81 and a standard deviation of 17.19 for a single mean estimate, at least 25 individuals needed to be included in the study [6]. The study was carried out with a total of 85 patients.

The study protocol was approved by the Ethics Committee of Istanbul Medeniyet University Health Sciences (2021/0150) and conducted in accordance with the Declaration of Helsinki. Written informed consent was obtained from all participants.

## Demographic data

Demographic data (age, gender), education level, employment, type of residence, marital status, number of persons in household, monthly income and source of income were questioned.

Number of persons in household was divided into 4 groups: (1) 1–2, (2) 3–4, (3) 5–7, (4) >8 persons.

Source of income was divided into 2 groups: (1) own salary, (2) family. The type of residence was divided into 2 groups: (1) living alone, (2) living with family.

## Disability-specific data

The cause, subtype and duration of disability were questioned. Disability causes were divided into 2 groups: (1) congenital and (2) acquired.

Disability subtype was divided into 3 groups: (1) neurological, (2) orthopedic, (3) other.

Other were classified as physical disabilities that can arise from internal or oncological problems, distinct from musculoskeletal or neurological conditions such as chemotherapy-related neuropathy, diabetic foot complications, severe gastrointestinal or renal diseases (such as inflammatory bowel disease, malabsorption syndromes, end-stage renal disease or dialysis dependence etc.).

Since the study was conducted at a Physical Medicine and Rehabilitation outpatient clinic by the Health Board for Persons with Disabilities, it focused on individuals with physical disabilities related to musculoskeletal or neurological conditions. Purely neurological conditions without musculoskeletal involvement (for example migrane, epilepsy, Alzheimer's disease etc.) were not included.

The duration of disability was divided into 3 groups: (1) 1–5 years, (2) 6–10 years, (3) >10 years.

## Mobility

Mobility of the participants was categorized into 3 groups as independently mobile, mobile with assistive device/help and wheelchair dependent.

## Personal wellbeing index

The Personal Wellbeing Index-Adult (PWI-A) scale is a thematic and 11-point Likert-type (0–10) tool that aims to measure subjective wellbeing through the satisfaction levels of individuals in eight areas of living (standard of living, personal health, achievements in life, personal relationships, personal safety, community-connectedness, future security, and spirituality/religion) in accordance with the structure of the concept. The total score obtained from the scale corresponds to the average of eight subdomains, and higher scores indicate greater satisfaction with life. Validity and reliability of the Turkish version of the scale have been demonstrated [7].

## Keele Assessment of Participation

The Keele Assessment of Participation (KAP) was developed by Ross Wilkie et al in 2005 [8]. KAP assesses participation in 11 aspects of life, based on the individual's own perceptions. The instrument has been designed as a generic measurement tool for use in adults in the general population. Responses are on a five-point ordinal scale (all, most, some, a little, none of the time). To date, studies calculated a total score by dichotomizing the categories into restricted (some, little, none) and not restricted (all, most) and simply counting the numbers of restrictions on the 11 items [8].

## Statistical analysis

Descriptive statistics were reported as numbers and percentages for categorical data, and as median, minimum and maximum for numerical data. Whether the numerical data followed a normal distribution was examined using the Shapiro-Wilk test. Pearson's chi-square and Fisher exact tests were used to analyze the relationships between the categorical variables. Mann-Whitney U test was used for comparison of means between the groups with two categories. The Friedman test was used for comparison of means groups with more than two categories, and multiple comparisons were made with Dunn's test. GLM models were used for multivariate analysis of the variables that were found to be significant as a result of univariate analyses. Statistical significance level was set at 0.05 and SPSS Statistics for Windows, version 26.0 (IBM Corp., Armonk, NY) was used for statistical analysis.

## Results and discussion

Descriptive statistics of the sociodemographic variables are presented in **Table 1** and disability-related variables in **Table 2**.

## Assessment of limitation of participation

**Table 3** shows the descriptive statistics and p values obtained following the analysis of KAP scores among the groups. From the table, it can be seen that there is no significant difference among the groups in terms of education level, cause of disability, type of residence, number of persons in household, and duration of disability as assessed by median KAP scores ($p = 0.242$; $p = 0.921$; $p = 0.161$; $p = 0.267$; $p = 0.999$, respectively). Females had a a significantly higher median KAP score than males ($p = 0.009$). The median KAP score of the single group was significantly higher than the married group ($p = 0.050$). The unemployed group showed a significantly higher median KAP score than the employed group ($p < 0.001$). The median KAP score of those with a neurologic disability was significantly higher compared to the other two groups ($p = 0.050$). Those living on family income showed a significantly higher median KAP score than those with earned income ($p < 0.001$). A significant difference was observed among the three groups with different levels of mobility in terms of median KAP scores ($p < 0.001$). When the differences among these 3 groups were examined in detail, the median KAP score of the independently mobile group was found to be significantly lower than those of other two groups ($p = 0.001$, $p < 0.001$). No significant difference was observed in median KAP scores between the group who required assistance/help and nonmobile group ($p = 0.170$).

## Factors affecting participation

**Table 4** shows the results of the multivariate analysis. The factors affecting KAP were determined as employment status, mobility level and personal wellbeing.

**Table 1. Sociodemographic characteristics of the participants.**

| | n (%), median (min-max) |
|---|---|
| **Gender, n(%)** | |
| Male | 42 (49,4) |
| Female | 43 (50,6) |
| **Age (years), median (min-max)** | 49.0 (18–65) |
| **Education status, n(%)** | |
| Illiterate | 18 (21,2) |
| Primary school | 37 (43,5) |
| High school | 23 (27,1) |
| University | 7 (8,2) |
| **Marital status, n(%)** | |
| Yes | 62 (72,9) |
| No | 23 (27,1) |
| **Working status, n(%)** | |
| Yes | 27 (31,8) |
| No | 58 (68,2) |
| **Income source, n(%)** | |
| Own salary | 42 (49,4) |
| Family | 43 (50,6) |
| **Monthly income (TL), median (min-max)** | 4500 (700–20000) |
| **Accomodation, n(%)** | |
| Alone | 8 (9,4) |
| With family | 77 (90,6) |
| **Household number, n(%)** | |
| 1–2 | 18 (21,2) |
| 3–4 | 50 (58,8) |
| 5–7 | 17 (20,0) |

## Factors affecting willingness to participate

The last 3 questions of the scale are conditional questions and it was questioned whether the participants were willing to participate in voluntary work, education/training courses, and social activities. The answers to these questions were used to evaluate "willingness to participate". Those who answered "yes" to all 3 questions were considered as willing to participate. No significant relationship was observed between willingness to participate and gender, marital status, education level, employment status, type of disability, cause of disability, number of persons in household, duration of disability, income source and personal wellbeing variables (p = 0.312; p = 0.478; p = 0.845; p = 0.732; p = 0.654; p = 0.838; p = 0.438; p = 0.383; p = 0.778; p = 0.110, respectively). The willingness to participate was significantly greater among those living with their families than those living alone (p = 0.030). Independently mobile subjects showed significantly greater willingness to participate compared to the other two groups (p = 0.039). There was no significant difference between those who were willing to participate and those who were not with respect to mean age and monthly income (p = 0.155; p = 0.134).

Disability is a condition that is associated with physical, psychological and social problems. While having a disability is defined as being outside of normality in the biological sense, in the social sense, it is defined as social and cultural limitations of an individual's ability to lead an independent life and to fulfill their desired roles in the society [9].

**Table 2. Disability specific characteristics of the participants.**

|  | n (%), median (min-max) |
|---|---|
| **Disability type, n(%)** |  |
| Congenital | 20 (23,5) |
| Acquired | 65 (76,5) |
| **Disability cause, n(%)** |  |
| Orthopedic | 26 (30,6) |
| Neurologic | 41 (48,2) |
| Other | 18 (21,2) |
|   Oncologic | 9 (50) |
|   Internal | 9 (50) |
| **Disability duration, n(%)** |  |
| 1–5 years | 41 (48,2) |
| 6–9 years | 14 (16,5) |
| ≥10 years | 30 (35,3) |
| **Mobility status, n(%)** |  |
| Independent | 45 (52,9) |
| Assistive device/help | 20 (23,5) |
| Wheelchair dependent | 20 (23,5) |
| **Personal Well-Being Index (PWI), median (min-max)** | 30 (10–76) |
| **Keele Assessment of Participation (KAP), median (min-max)** | 7 (1–11) |
| **KAP—willing to participate, n(%)** |  |
| Yes | 74 (87,1) |
| No | 11 (12,9) |

Social integration, namely participation can be defined as involvement in life situations and denotes a person's performance of tasks and actions in their actual environment including activities with their community, family, peers or friends [10].

Participation has been one of the important issues that have been focused on in recent years.

Therefore, in this study, the determinants of participation in individuals with disabilities resulting from different causes and with different mobility levels were investigated.

There are currently 2,511,950 people living with a disability in Turkey who are registered on the Turkish National Disability Database, of whom 1,414,643 are men and 1,097,307 are women (56% men, 44% women). Among them, 40.63% have chronic disease and 13.8% have orthopedic disability. According to the latest data from the Turkish Statistical Institute, 6.5% of individuals over the age of 15 cannot walk (8.9% for women, 4% for men), and 8.7% cannot go up and down stairs (12.4% for women, 5% for men) [11].

The working age group, 18 to 65 years of age, which is the age range of people who generally work in paid or unpaid jobs, was included in the study. According to the Population and Housing Survey conducted in 2011, the labor force participation rate of the population with at least one disability is 35.4% for men, 12.5% for women, and 22.1% in total [11]. In this study, the labor force participation rates were 42.9% for men and 20.9% for women and 27% in total, which are higher than the latest population data. While the mean age of people with a physical disability in Turkey was reported as 33.8 years in the same survey [11], the participants included in this study were older (mean age 49 years). This result was obtained despite the higher mean age of the participants, suggesting that the employment of people with disabilities has increased gradually.

**Table 3. Evaluation of Keele Assessment of Participation.**

|  |  | n | Mean | Median | SD | Min | Max | p |
|---|---|---|---|---|---|---|---|---|
| **Gender** | female | 43 | 7.60 | 8 | 2.09 | 2.00 | 11.00 | 0.009 |
|  | male | 42 | 6.21 | 7 | 2.67 | 1.00 | 10.00 |  |
| **Marital status** | no | 23 | 7.60 | 8 | 2.42 | 1.00 | 10.00 | 0.050 |
|  | yes | 62 | 6.66 | 7 | 2.47 | 1.00 | 11.00 |  |
| **Education status** | Illiterate | 18 | 7.50 | 8 | 2.09 | 3.00 | 10.00 | 0.242 |
|  | primary school | 37 | 7.21 | 8 | 2.29 | 1.00 | 11.00 |  |
|  | high school—university | 30 | 6.20 | 7 | 2.80 | 1.00 | 10.00 |  |
| **Working status** | no | 58 | 7.82 | 8 | 1.86 | 1.00 | 11.00 | <0.001 |
|  | yes | 27 | 4.96 | 5 | 2.54 | 1.00 | 9.00 |  |
| **Disability cause** | congenital | 20 | 7.00 | 7 | 2.15 | 3.00 | 10.00 | 0.921 |
|  | acquired | 65 | 6.89 | 7 | 2.59 | 1.00 | 11.00 |  |
| **Disability type** | orthopedic | 26 | 6.53 | 7 | 2.43 | 1.00 | 10.00 | 0.050 |
|  | neurologic | 41 | 7.60 | 8 | 1.96 | 2.00 | 10.00 |  |
|  | other | 18 | 5.88 | 7 | 3.19 | 1.00 | 11.00 |  |
| **Accomodation** | with family | 77 | 7.06 | 7 | 2.40 | 1.00 | 11.00 | 0.161 |
|  | alone | 8 | 5.50 | 6 | 2.97 | 1.00 | 9.00 |  |
| **Household number** | 1–2 | 18 | 6.11 | 7 | 2.63 | 1.00 | 10.00 | 0.267 |
|  | 3–4 | 50 | 7.14 | 8 | 2.33 | 1.00 | 10.00 |  |
|  | 5–7 | 17 | 7.11 | 7 | 2.71 | 2.00 | 11.00 |  |
| **Disability duration** | 1–5 years | 41 | 6.87 | 7 | 2.69 | 1.00 | 11.00 | 0.999 |
|  | 6–9 years | 14 | 6.71 | 8 | 2.89 | 1.00 | 10.00 |  |
|  | >10 years | 30 | 7.06 | 7 | 2.01 | 3.00 | 10.00 |  |
| **Income source** | own salary | 42 | 5.80 | 7 | 2.70 | 1.00 | 10.00 | <0.001 |
|  | family | 43 | 8.00 | 8 | 1.66 | 3.00 | 11.00 |  |
| **Mobility status** | independent | 45 | 5.64 | 6 | 2.53 | 1.00 | 10.00 | <0.001 |
|  | assistive device / help | 20 | 8.00 | 8 | 1.77 | 3.00 | 11.00 |  |
|  | wheelchair | 20 | 8.70 | 8 | .92 | 7.00 | 10.00 |  |

KAP was not significantly associated with age and monthly income (p = 0.382, p = 0.759).

When the unemployed individuals were analyzed according to the cause of disability, unemployment rate was highest in those with a neurological disability (n = 34, 82.9%), while nearly half of the individuals with an orthopedic disability were unemployed. The unemployment rate was 90% in wheelchair-dependent individuals, of whom only 2 were employed.

Understanding and addressing the unique work barriers faced by people with disabilities is crucial for creating inclusive and supportive work environments that promote their employment opportunities and job retention. Kusznir Vitturi et al. [12] provides a detailed systematic review of the work barriers and job adjustments experienced by people with Multiple Sclerosis (MS), highlighting the diverse challenges faced by individuals with MS in the workplace, including job characteristics, work environment factors, social relationships, negative work events, and emphasizes the importance of tailored job adjustments, such as workload management, flexible work schedules, workplace adaptations, and vocational rehabilitation services, to promote better work outcomes. Also in another review Kusznir Vitturi et al. [13] conducted a comprehensive analysis of the potential impact of various factors, including demographic variables, disease severity, and types of work barriers, on the occupational outcomes of people with MS. It discusses the importance of identifying and addressing specific work barriers, utilizing appropriate job adjustments, and promoting supportive workplace environments to

Table 4. Multivariate analysis results showing factors affecting Keele Assessment of Participation.

| Parameter | B | p | 95% Confidence Interval | |
|---|---|---|---|---|
| | | | Lower Bound | Upper Bound |
| Intercept | 7,155 | <0,001 | 5,256 | 9,054 |
| female | ,777 | ,071 | -,068 | 1,623 |
| male | | | | reference |
| [marital status =, no | -,359 | ,477 | -1,357 | ,640 |
| [marital status = yes | | | | reference |
| [working status = no | 1,912 | ,002 | ,703 | 3,122 |
| [working status = yes | | | | reference |
| [disability type = orthopaedic] | ,883 | ,145 | -,312 | 2,077 |
| [disability type = neurologic] | ,433 | ,463 | -,736 | 1,602 |
| [disability type = other | | | | reference |
| [income source = own] | -,111 | ,850 | -1,277 | 1,055 |
| [income source = family] | | | | reference |
| [mobility status = independent] | -1,508 | ,024 | -2,809 | -,207 |
| [mobility status = assistive device / help] | -,034 | ,958 | -1,293 | 1,225 |
| [mobility status = wheelchair] | | | | reference |
| Personal wellbeing index | -,046 | ,050 | -,093 | ,000 |

enhance the employment prospects and job retention of individuals with MS. These articles underscores the crucial role of effective communication between healthcare professionals, patients, employers, and colleagues in addressing these barriers and implementing successful job adjustments.

The majority of the individuals participating in the study lived with their families. Although half of the individuals were independently mobile and 27% were single, the number of people living alone was low, reflecting the strong family ties of the Turkish society.

Noreau and Fougeyrollas reported low social participation in persons with long-term disabilities, and these difficulties increase with advancing age [14]. Similarly, in a socio-demographic research study on people with disabilities, Cardol et al. stated that age affects participation negatively [4]. Bodde et al., on the other hand, reported that while most studies show an association between participation and aging, participation is not associated with age, but with better health status [15]. Although the mean age of our subjects was similar to those reported by other studies, no relationship was found between age and participation, which is in line with Bodde et al.'s findings. As the health status of the subjects included in our study did not differ among age groups, participation rates were comparable.

In this study, limitations in social participation were more pronounced among women and single subjects. Cardol et al. examined social participation of people with chronic disabilities and showed that gender is an effective factor in participation, and that social participation of women living with chronic conditions such as rheumatoid arthritis for a long time are more negatively affected than men. Barf et al. investigated the factors limiting social participation in young adults with spina bifida and showed that 71% of individuals did not have a partner and this negatively affected their social participation [16].

Social participation restriction was also found to be related to employment, and making a living on one's own. Cardol et al. also have reported that working positively affects social participation of an individual with a disability [4]. Consistently, Ostir et al. reported that going to work positively affects social participation [17]. It is considered that commuting to work is participation in itself, and working in a job will affect social participation of the individual

since it will contribute to the sense of belonging to the society, being respected and being productive in a positive way. In this context, prejudices about the disabled people should be eliminated with a community-based approach, support and services should be provided, physical environments and workplaces should accommodate the needs of disabled persons, and policies that encourage employment of people with disabilities should be followed.

While congenital or acquired disability did not affect participation, participation restriction was more pronounced in those with a neurological disability compared to the other two groups. Likewise, Ostir et al. reported that individuals with orthopedic impairments were less affected by participation restriction than individuals with neurological disorders [17].

Social participation was more pronounced in those who were independently mobile than those who needed assistance/help or were wheelchair dependent. In a study, Chaves et al. stated that the barriers and the environment affect participation and showed that the use of a wheelchair will not provide the person with the opportunity to move if the environment is not suitable [18]. Barf et al. examined the factors affecting participation of young adults with spina bifida and found that wheelchair dependence was negatively associated with participation [16]. The finding of our study that wheelchair users experience restrictions in participation in social life and have reduced employment opportunities is consistent with literature reports. The high cost of technological aids, physical conditions in public and private institutions, as well as negative attitudes and prejudices in the society can negatively affect social participation of persons with disabilities. In order to ensure accessibility for all, it is important to improve access to public transport and make further arrangements to create a barrier-free physical environment for the disabled and to increase social awareness through community-based rehabilitation.

There are many studies in the literature reporting the relationship between quality of life and participation in individuals with disability [19–23]. Personal wellbeing, which is used as a general term to describe how an individual feels about his/her own life, includes emotional experiences of individuals, their satisfaction with life, and the subjective perception of quality of life in general [24]. In this context, personal wellbeing is measured by the level of satisfaction with what an individual feels about themselves. In this study, median PWI-A scores of the participants were well below the reference values [25, 26], indicating poor wellbeing. Also, personal wellbeing has been found as one of the determinants of participation. Wolman et al. reported that wellbeing scores were lower in adolescents with chronic conditions, and having a disability was associated with wellbeing [27]. Di Cagno et al. also reported the relationship between social participation and wellbeing in people with vision impairment [28].

Although social participation of the study sample was low, the willingness to participate was high. Even in the wheelchair-dependent group, the willingness to participate was 70%. This suggests that social participation of disabled persons can be greatly increased with the right actions to be implemented at the individual and community levels. In the current study, it was found that the groups living with their families and those who are independently mobile were more willing to participate.

The results of the study both confirmed some existing expectations and brought forth surprising findings, offering valuable insights for healthcare professionals in assisting individuals with disabilities in clinical practice.

Some of the expected results include the relationship between employment and social participation, where being employed positively affected social participation. This aligns with previous studies that have consistently shown how working positively influences an individual's sense of belonging, respect, productivity, and overall social integration [4, 17]. Additionally, the correlation between personal well-being and participation was also anticipated, as higher

levels of well-being have been associated with increased social engagement and satisfaction with life.

On the other hand, some results were unexpected and shed new light on the dynamics of social participation among individuals with disabilities. For instance, while age has often been considered a factor affecting participation, this study found no significant relationship between age and participation. This surprising finding challenges previous assumptions and suggests that other factors such as health status and social support systems may play a more critical role in determining social participation among different age groups of individuals with disabilities.

Moreover, the study revealed that social participation was more pronounced among those living with their families and those who were independently mobile. This emphasizes the importance of family support and accessible environments in facilitating social engagement for individuals with disabilities. The high willingness to participate across different disability groups, especially among wheelchair-dependent individuals, highlights the untapped potential for increased social participation with targeted interventions and community-based support systems.

These new findings can significantly aid healthcare professionals in clinical practice by guiding them to prioritize certain areas when assisting individuals with disabilities. For example, focusing on promoting employment opportunities, creating accessible environments, providing social support networks, and addressing personal well-being can all contribute to enhancing social participation and overall well-being for individuals with disabilities. Healthcare professionals can use these insights to tailor interventions, advocacy efforts, and policy recommendations that promote inclusivity, accessibility, and empowerment for individuals with disabilities in their communities.

## Study limitations

A number of limitations should be noted for our study.

There are several potential biases that need to be considered as methodological limitations.

Firstly, selection bias could have influenced the study results. The participants were recruited from a physical medicine and rehabilitation outpatient clinic by the Health Board for Persons with Disabilities, which may not represent the entire population of individuals with disabilities. Those who seek healthcare services may have different characteristics, needs, and levels of social participation compared to individuals who do not regularly visit healthcare facilities. The study setting was a university hospital in the city center and the subjects presenting to the hospital were living at the city center. Therefore, people with disabilities living in rural areas were not included in our study. Therefore, the sample may not be fully representative of all individuals with disabilities.

Secondly, information bias could have occurred due to the methods used to collect data. The study relied on self-reported measures for variables such as mobility level, employment status, and personal well-being. Self-reported data are susceptible to recall bias and social desirability bias, where participants may provide responses that they perceive as socially acceptable or that align with their desired self-image. This could potentially lead to overestimation or underestimation of certain factors, affecting the accuracy of the results.

Another limitation is that environmental factors have not been evaluated in detail. The effect of social environment, attitudes and assistive technologies on participation could not be examined.

Moreover, the study's cross-sectional design limits the ability to establish causality between variables. Longitudinal studies would provide more robust evidence regarding the relationships between demographic, disability-related, and participation-related factors over time.

Additionally, the study's sample size of 85 participants may limit the generalizability of the findings to larger populations of individuals with disabilities.

Despite these limitations, the study contributes valuable insights into the factors associated with social participation among individuals with disabilities. Future research should aim to address these methodological limitations by employing larger and more diverse samples, using objective measures in addition to self-report measures, and utilizing longitudinal designs to establish causal relationships and track changes over time.

## Conclusions

In conclusion, the results showed that mobility level, employment status and personal wellbeing are the determinants of social participation in people with disabilities. In particular, participation was significantly lower in individuals with mobility limitations and individuals with neurological disabilities. These findings may serve as a guide for community-based rehabilitation studies and for the development of social policies for people with disability in Turkey.

## Supporting information

**S1 Text. Patient evaluation form.**
(PDF)

**S1 Dataset.**
(XLSX)

## Author Contributions

**Conceptualization:** Tugce Pasin, Bilinc Dogruoz Karatekin.

**Data curation:** Bilinc Dogruoz Karatekin.

**Formal analysis:** Bilinc Dogruoz Karatekin.

**Investigation:** Tugce Pasin, Bilinc Dogruoz Karatekin.

**Methodology:** Tugce Pasin, Bilinc Dogruoz Karatekin.

**Resources:** Bilinc Dogruoz Karatekin.

**Software:** Bilinc Dogruoz Karatekin.

**Supervision:** Tugce Pasin.

**Visualization:** Bilinc Dogruoz Karatekin.

**Writing – original draft:** Tugce Pasin, Bilinc Dogruoz Karatekin.

**Writing – review & editing:** Tugce Pasin, Bilinc Dogruoz Karatekin.

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
