## [Decision Letter · Decision Letter 0]

4 Apr 2024

PONE-D-24-02938Determinants of social participation in people with disabilityPLOS ONE

Dear Dr. Dogruoz Karatekin,

Thank you for submitting your manuscript to PLOS ONE. After careful consideration, we feel that it has merit but does not fully meet PLOS ONE’s publication criteria as it currently stands. Therefore, we invite you to submit a revised version of the manuscript that addresses the points raised during the review process.

Please go through the reviewer's comments, make proper corrections, and submit the revised version.

We look forward to receiving your revised manuscript.

Kind regards,

Md. Feroz Kabir, BPT, MPT, MPH, BPED, MPED

Academic Editor

PLOS ONE

Journal Requirements:

Additional Editor Comments:

Please go through the reviewers comments and do proper correction and submit the revised version.

Reviewers' comments:

Reviewer's Responses to Questions

**Comments to the Author**

1. Is the manuscript technically sound, and do the data support the conclusions?

Reviewer #1: Partly

Reviewer #2: Yes

Reviewer #3: Partly

2. Has the statistical analysis been performed appropriately and rigorously? 

Reviewer #1: No

Reviewer #2: Yes

Reviewer #3: No

3. Have the authors made all data underlying the findings in their manuscript fully available?

Reviewer #1: No

Reviewer #2: Yes

Reviewer #3: Yes

4. Is the manuscript presented in an intelligible fashion and written in standard English?

Reviewer #1: Yes

Reviewer #2: Yes

Reviewer #3: Yes

5. Review Comments to the Author

Reviewer #1: I reviewed and examined the paper and what the authors have prepared- it has been established and also part of the norms. Individuals with disabilities certainly will not be able to be mobile or have less social participations. I do not see how the manuscript brings new idea/knowledge to the field.

Reviewer #2: Dear Authors,

I had the pleasure of reading your manusript. The article is interesting and important but there are many issues that need to clarified and included before an eventual publication.

1) The authors need to provide more information on how the research subjects were enrolled. Were they all consecutive patients? Did the authors set any exclusion criteria?

2) I suggest the authors specifiy the type of disabilities the research subjects had. Did the authors included any subject with mental or cognitive disability? The definiton of disability together with the reference used should be included in this part of the manuscript as well.

3) "Since the patients who presented to the Physical Medicine and Rehabilitation outpatient clinic were included in this study, patients without musculoskeletal problems could not be evaluated. " - This sentnece is confusing and may indicate an important source of bias in the study. What do the authors understand with "musculoskeletal problems". So all the research subjects had an orthopedic cause of disability at least? Patients with a pure neurological disease were not included in the study?

4)In many parts of the "Results", there are some mistakes i the text "Error! Reference source not found".

5) The authors should provide more information on what they consider as "other" types of disability.

6) Did the authors assess the degree of the disability somehow?

7) Employment/income had the mopst important statistical significant results. Therefore, the discussion must be improved with other arguments that may explain this observation. People with disabilities face work barriers, require several job adjustments and are also vulnerable to stigma and discrimiation at work. Please see and cite doi: 10.3390/ijerph19159452 and doi: 10.1007/s10926-022-10084-1.

Reviewer #3: I appreciate the opportunity to review this study. It addresses a relevant topic; however, I have noted some significant considerations regarding the methods employed, which may impact the study findings. I am particularly concerned about the limitations of the cross-sectional design used, as well as the relatively small sample size. Below, I outline some points that deserve attention for the improvement of the work:

The abstract lacks detail regarding the sample characteristics and the number of participants, which are crucial elements for understanding and interpreting the results.

In the introduction, I suggest expanding the theoretical framework, discussing existing knowledge on factors associated with participation, and identifying gaps in the literature. It is essential to justify how this study will contribute to advancing current knowledge on the topic.

Regarding the methods, I emphasize that the cross-sectional design used in the study has limitations. While a cross-sectional study examines the relationship between exposure and outcome variables at a single point in time, providing an instantaneous view of associations in the population, it also has limitations. It does not establish causality, as the temporal order of variables cannot be determined. There may be selection bias if the sample is not representative, compromising the generalizability of the results. Additionally, information bias may arise if participants provide incorrect information about their exposures or outcomes. I recommend a review of the study objectives in light of these considerations, as well as a clear explanation of the variables investigated, including the outcome variable and predictors.

Another point to consider is the sample size. It is essential that the sample size is properly justified, either through sample size calculation or based on relevant references. Additionally, I suggest a more detailed analysis of the characteristics of participants' disabilities, considering the possibility of different repercussions depending on the type and severity of the disability.

Regarding the results, there appear to be writing problems indicated by the word "Error!", which require correction by the authors.

Finally, in the discussion section, I recommend a more critical and reflective approach to the results, contextualizing them in relation to existing knowledge and discussing their practical implications for the field of health and rehabilitation. The discussion is overly descriptive; the first two paragraphs provide context but do not discuss the results. It is essential in the discussion to contrast the findings with existing knowledge or justify the results. The authors should highlight whether the results were expected or surprising and discuss how the new findings can aid healthcare professionals in the clinical practice of assisting individuals with disabilities.

6. PLOS authors have the option to publish the peer review history of their article (what does this mean?). If published, this will include your full peer review and any attached files.

Reviewer #1: No

Reviewer #2: No

Reviewer #3: **Yes: **Soraia Micaela Silva

---

## [Author Response · Author response to Decision Letter 0]

23 Apr 2024

Editor - inChief

Thank you for your kind review.

The headings of the manual were edited according to PLOS ONE's style requirements. 

The full ethics statement and the name of the ethics committee have been added.

Reviewer 1

Thank you for your kind review.

This manuscript makes a valuable contribution to the literature on disability studies and social participation. By investigating the determinants of social participation in people with disabilities, specifically focusing on mobility levels, employment status, and personal wellbeing, the study provides insights that can guide community-based rehabilitation efforts and the development of social policies. The findings emphasize the importance of addressing mobility limitations, promoting employment opportunities, and enhancing personal wellbeing to facilitate greater social participation among individuals with disabilities. Additionally, the study sheds light on gender disparities, highlighting the higher social participation levels among females, as well as the impact of neurological disabilities on social integration. Overall, this research adds depth to our understanding of the factors influencing social participation in the disabled population and underscores the need for targeted interventions and inclusive policies to improve their quality of life and societal engagement.

Reviewer 2

Thank you for your kind review.

1) Regarding the enrollment of research subjects, they were indeed consecutive patients who visited the Physical Medicine and Rehabilitation outpatient clinic by the Health Board for Persons with Disabilities within the specified timeframe. However, we omitted to mention the exclusion criteria in the manuscript. The exclusion criteria included individuals with mental or cognitive disabilities due to the focus on physical disabilities and their impact on social participation. We included this information in the revised manuscript to provide clarity on subject selection.

2) The types of disabilities among the research subjects were primarily physical in nature, including neurological and orthopedic disabilities. We did not include subjects with mental or cognitive disabilities, as mentioned earlier. We specified these details in the manuscript along with the definition of disability as per the World Health Organization's International Classification of Functioning, Disability, and Health (ICF).

3) The sentence regarding musculoskeletal problems indeed needs clarification. It means that since the study was conducted at a Physical Medicine and Rehabilitation outpatient clinic by the Health Board for Persons with Disabilities, it focused on individuals with physical disabilities related to musculoskeletal or neurological conditions. Purely neurological conditions without musculoskeletal involvement were not included. We rephrased this sentence to avoid confusion and provide a clearer explanation of the inclusion criteria.

4) We apologize for the errors in the text, and we ensured to correct these and provide accurate references in the revised manuscript.

5) We elaborated on the "other" types of disability in the manuscript, specifying examples and providing more information to enhance clarity.

6) The degree of disability was not assessed in this study. However, we acknowledge that assessing the severity or degree of disability could provide valuable insights, and we will consider this for future research.

7) We appreciate the suggestion to improve the discussion section regarding employment and income as significant factors. We included additional arguments and cite the suggested references to provide a more comprehensive discussion on the barriers faced by people with disabilities in the workforce and how they impact social participation.

Thank you for the constructive feedback, and we addressed these points in the revised manuscript to enhance the quality and clarity of our research.

Reviewer 3

1) The abstract revised to include more detail about the sample characteristics and the number of participants, ensuring that readers can better understand and interpret the results.

2) In the introduction, we expanded the theoretical framework by discussing existing knowledge on factors associated with participation, identifying gaps in the literature, and clearly justifying how this study will contribute to advancing current knowledge on the topic.

3) Regarding the methods, we acknowledge the limitations of the cross-sectional design used in the study. We reviewed the study objectives in light of these considerations and provide a clear explanation of the variables investigated, including the outcome variable and predictors. Additionally, we addressed the potential for selection bias and information bias in the discussion of methodological limitations.

4) The sample size is properly justified either through sample size calculation or based on relevant references. 

5) Writing problems indicated by the word "Error!" in the results section is corrected promptly to ensure the accuracy and readability of the manuscript.

6) In the discussion section, we adopted a more critical and reflective approach to the results. This involved contrasting the findings with existing knowledge, discussing their practical implications for the field of health and rehabilitation, and highlighting whether the results were expected or surprising. We also discussed how the new findings can aid healthcare professionals in the clinical practice of assisting individuals with disabilities.

Thank you for the valuable feedback, and we incorporated these suggestions to improve the overall quality and impact of our study.

---

## [Editor Report · Decision Letter 1]

3 May 2024

Determinants of social participation in people with disability

PONE-D-24-02938R1

Dear Bilinc Dogruoz Karatekin,

We’re pleased to inform you that your manuscript has been judged scientifically suitable for publication and will be formally accepted for publication once it meets all outstanding technical requirements.

Kind regards,

Md. Feroz Kabir, BPT, MPT, MPH, BPED, MPED

Academic Editor

PLOS ONE

Additional Editor Comments (optional):

Please submit the revised manuscript with the improvements to the English grammar within the next 15 days.
---

## [Editor Report · Acceptance letter]

8 May 2024

PONE-D-24-02938R1 

PLOS ONE

Dear Dr. Dogruoz Karatekin, 

I'm pleased to inform you that your manuscript has been deemed suitable for publication in PLOS ONE. Congratulations! Your manuscript is now being handed over to our production team.

Kind regards, 

on behalf of

Dr. Md. Feroz Kabir 

Academic Editor

PLOS ONE